# Sexual Satisfaction and Quality of Life in Cardiovascular Patients: The Mediating Role of Anxiety

**DOI:** 10.3390/healthcare11030290

**Published:** 2023-01-18

**Authors:** Maria Stella Epifanio, Sabina La Grutta, Pietro Alfano, Salvatore Marcantonio, Marco Andrea Piombo, Martina Ammirata, Eduardo Rebulla, Silvia Grassi, Simona Leone, Francesco Clemenza, Rosario Girgenti, Rosa Lo Baido, Maria Di Blasi

**Affiliations:** 1Department of Psychology, Educational Science and Human Movement, University of Palermo, 90128 Palermo, Italy; 2Institute of Translational Pharmacology (IFT), National Research Council of Italy, 90146 Palermo, Italy; 3Quality, Planning and Strategic Support Area, University of Palermo, Piazza Marina 61, 90133 Palermo, Italy; 4Department of Psychology “Renzo Canestrari”, Alma Mater Studiorum, University of Bologna, 40127 Bologna, Italy; 5Division of Cardiology, Candela Clinic, 90141 Palermo, Italy; 6Cardiology Operating Unit, IRCCS-ISMETT (Istituto Mediterraneo per i Trapianti e Terapie ad Alta Specializzazione), 90127 Palermo, Italy; 7Clinical Psychology Service, IRCCS-ISMETT (Istituto Mediterraneo per i Trapianti e Terapie ad Alta Specializzazione), 90127 Palermo, Italy; 8Section of Psychiatry, Experimental Biomedicine, Clinical Neuroscience and Advanced Diagnostic Department (BiND), Palermo University, 90127 Palermo, Italy

**Keywords:** sexual satisfaction, quality of life, anxiety

## Abstract

Background: Cardiovascular diseases represent one of the most important problems for public health. Research indicates that elderly patients consider sexual satisfaction as a fundamental aspect of their quality of life and a better sexual function is related to higher general wellbeing. Objective: The main objective of this study was to investigate the mediating role of anxiety and depression in the relationship between quality of life and sexual satisfaction in cardiovascular (CVD) patients. Methods: The sample comprised 128 adult patients, males and females, hospitalized in a Cardiology Rehabilitation clinical center. To collect data, the following were used: a demographic information sheet, the left ventricular ejection fraction (EF) to evaluate cardiac function, cardiovascular diagnosis type, the HADS scale to evaluate anxiety and depression states, a test for sexual satisfaction evaluation (SAS) and the SF-36 survey to measure quality of life. Results: The results indicated that only SF-36 physical health is indirectly related to SAS through its relationship with anxiety. Conclusion: A mediating model was proposed to explore the underlying association between sexual satisfaction and quality of life. We recommend investigating perceived general health and sexuality as clinical indicators for therapeutic decisions and risk evaluation for the management of cardiovascular diseases.

## 1. Introduction

Cardiovascular diseases are the most common causes of death according to European society Cardiology (ESC) member countries, with 2.2 million deaths in females and 1.9 million deaths in males. Ischemic heart disease is the leading cause of death among cardiovascular diseases in both females and males at 38% and 44%, respectively. Stroke is the second most common cause of CVD deaths, accounting for 26% of all deaths in females and 21% in males, respectively [1] 

The literature shows an increasing interest in sexual behaviours of people with cardiovascular disease, probably due to the impact on health status [2,3]. Nevertheless, research is still scarce regarding the health consequences on sexual life for elderly adults [4]. In CVD, there are concerns about how a sex life might affect a patient’s heart condition [5]. The recent literature argues that sexuality brings uniform health benefits in older adults [6], and moreover, an inverse association has been observed between reported frequency of sexual activity and mortality [3].

Research has evidenced many factors that can negatively influence sexual activity and satisfaction among heart failure (HF) patients. Pharmacotherapy can have a negative impact on sexual activity among HF patients. Specifically, sexual dysfunction is associated with the use of specific drug therapies such as beta-blockers, diuretics and cardiac glycoside drugs [7]. Sexual dysfunction was also observed in common comorbidities such as obesity and diabetes, chronic obstructive pulmonary disease (COPD) and hypertension [8].

Common symptoms of cardiovascular diseases such as dyspnoea, fatigue and difficulty in physical activity are also related to sexual problems [9] and are frequently obstacles to carrying out sexual activity [10].

Dysfunctions in the sexual sphere reduce health, longevity and satisfaction of life, as well as the development of couples and their intimate relations [11]. Satisfaction with one’s sexual activity represents a key feature in an intimate relationship, and thus the presence of sexual difficulties could lead toward stress and interpersonal conflicts, contributing to a reduction in the quality of life [5]. A recent review showed a positive relationship between sexual satisfaction and some factors such as emotional closeness, intimate communication, quality, satisfying relationships, psychological health, general sense of wellbeing, happiness and quality of life [8]. Satisfaction in sexual activity is considered important by most cardiovascular patients, including elderly patients, and for this reason, it could be suitable for health care professionals to provide sexual counselling for cardiac patients to improve their wellbeing [12]. Indeed, a reduced health-related quality of life (HRQoL) has been associated with experiencing sexual problems [8].

HRQoL is a critical factor in the management of patients with cardiovascular diseases and has been indicated as an important index of an individual’s general wellbeing. Health-related quality of life can provide information both on how patients perceive their illness and how it affects physical, mental, emotional and social functioning [13].

Moreover, HRQOL assessment may be useful to identify individuals at an increased risk of readmission and mortality in CVD [14].

According to Strick et al. [15], anxiety and depression were associated with more frequent visits to the outpatient cardiac clinic, re-hospitalizations due to cardiac events and higher medical expenditures. A key role of anxiety and depression in cardiovascular diseases has been highlighted by several studies in which both seem to be negative predictors of subsequent cardiac events [16,17].

Furthermore, in a recent study, the relationship between decreased sexual activity after a cardiac event was linked to patients’ reduced quality of life, psychological wellbeing and strain on intimate relationships [18]. Another study using a structural analysis evidenced that depressive symptoms mediated the association between sexual functioning and quality of life [19]. Moreover, Kriston et al. [20] found that depressive symptoms mediated the association between sexual function and quality of life in cardiovascular patients.

Among patients with cardiovascular disease, functional limitations and patient-rated health were associated with anxiety and depression [21]. In addition, decreased motivation toward sexual activity can reduce frequency without affecting the quality of sexual activity since the latter is related to fear of sexual activity, while sexual satisfaction is not affected [22].

Sexual satisfaction is associated with health-related quality of life for both men and women with CVD. This may play a different role depending on gender, because in men, cardiovascular risk factors play a stronger role while in women other factors are more important than sexual functioning [23].

Moreover, sexual satisfaction is an important factor in psychological and physical wellbeing [24].

Based on the above literature, this study aimed to explore the effects of sexual satisfaction on quality of life. In particular, anxiety and depression are known risk factors for decreased quality of life among patients with CVD [17] and could be the mediating factors in this relationship

We proposed the following two hypotheses:

**Hypothesis** **1.**
*Anxiety mediates the association between sexual satisfaction and quality of life in its physical (PCS) and mental component (MCS);*


**Hypothesis** **2.**
*Depression mediates the association between sexual satisfaction and quality of life in its physical (PCS) and mental component (MCS).*


## 2. Materials and Methods

### 2.1. Patients

We recruited 200 patients between April and March 2019 in a cardiovascular rehabilitation clinic. Participants attended a screening visit to (1) receive information about the research; (2) verify inclusion and exclusion criteria including being able to speak and read procedures; (3) collect information on general and socio-demographic characteristics. Subjects were enrolled if they met the following criteria: (a) age 18 years or older; (b) admission for specialized cardiovascular rehabilitation including patients who had not undergone a revascularization procedure, those who had a heart failure intervention and/or surgical revascularization diagnosed as having had heart failure at least 6 months prior; (c) no chronic condition that would limit life expectancy to less than 1 year; (d) absence of dementia and psychiatric problems. Subjects who experienced cognitive impairment or those who observed an oncological diagnosis were excluded. Of the 200 recruited subjects, 12 failed to meet the inclusion criteria, while 60 were excluded because they refused to complete the Sexual Adjustment Scale (SAS) of the Psychosocial Adjustment Scale [25] for the following reported reasons: 43 because they were widowed, 7 because they had prostate problems, 6 because they were single, and 4 because they refused to answer without specific motivation. The remaining sample comprised 128 patients with cardiovascular disease (89 men and 39 women, mean age 69.0 ± 9.3 years). Clinical characteristics were obtained from the patients’ medical records and included diabetes, hypertension, renal failure, left ventricular ejection fraction (FE) and cardiovascular diagnosis.

In this study, data collection was carried out after obtaining ethics and clinical approval from the institutional review board of the cardiovascular rehabilitation clinic (02/2019). This study included assuring the confidentiality of the data according to General Data Protection Regulation (EU GDPR). In addition, this study, which follows ethical requirements with medical research codes, was conducted according to the Principles of Good Clinical Practice (GCP) and the Declaration of Helsinki was observed in this study. All participants signed a written informed consent form before completing the questionnaire.

### 2.2. Measures

In order to assess HRQoL, sexual satisfaction, anxiety and depression, three self-administered questionnaires were administered, after obtaining the necessary permission and licences. To assess HRQoL, the Medical Outcomes Study 36-item Short Form Survey (SF-36) was used [26]. This is a questionnaire consisting of 36 items, structured in eight summary scales: physical functioning (PF), role physical limitation (RP), bodily pain (BP), general health perception (GH), vitality (VT), social functioning (SF), role limitation attributable both to emotional problems (RE) and mental health (MH). The PF, RP and BP scales were considered as the primary items of the physical component summary (PCS) score. SF, RE and MH were considered as the primary items of the mental component summary (MCS) score. GH and VT were considered as items of both dimensions. Internal consistency proved to be satisfactory and was (Cronbach’s alpha) a = 0.767, 0.72 for the PCS score and 0.75 for the MCS score in this study.

Data on sexual activity and satisfaction were collected using the SAS. This is a six-item self-report subscale of the Psychosocial Adjustment Scale (PAIS), designed to evaluate illness-related changes in the quality of relationships and sexuality [26]. The SAS is structured into six items that assess intimacy and discussions for the quality of relationships, sexual interest, frequency of sexual activity, pleasure and satisfaction and limitations for quality of sexual activity. Answers are scored by a four-point Likert scale ranging from 0 (no disturbance) to 3 (marked disturbance). The total score ranges from 0 to 18, with a higher score indicating more disturbance experienced. In this study, internal consistency proved to be satisfactory and was (Cronbach’s alpha) a = 0.78. Anxiety and depression were measured with the Hospital Anxiety and Depression Scale (HADS) [27]. The total scores for anxiety and depression range from 0 to 21; a higher score indicates higher levels of symptomatology. On either the anxiety or depression subscale, a score of 7 or less is considered normal, a score of 8-10 suggests a risk of psychological distress and 11 or higher reveals psychological distress. In this study, internal consistency proved to be satisfactory and was (Cronbach’s alpha) a = 0.84

### 2.3. Data Analyses

All statistical analyses were performed using SPSS version 20 statistical software. The sum scores for HADS, SF-36 and the average score for SAS were calculated in accordance with the questionnaire guidelines. Frequencies and percentages were used to describe the socio-demographic and medical data. A correlation analysis was used to analyze the relationships between the study variables. The mediation hypotheses were tested with a regression analysis using the SPSS macro software entitled PROCESS [28]. This data analysis procedure uses bootstrap sampling to estimate path coefficients for the regression equations derived from the hypotheses. A boot-strapping procedure (with 5000 bootstrap samples) was performed to estimate the 95% confidence interval for the indirect (mediated) effect [29]. The bootstrapping procedure does not perform the normality assumption on the sample distribution of indirect effects. It maintains a high power while providing adequate control over the type I error rate [29]. The test is statistically significant (at 0.05) if both confidence limits have the same sign. Zero is not a probable value so the null hypothesis of a null indirect effect must be rejected.

## 3. Results

A total of 128 patients were included in this study. The sample consisted mainly of male patients (69.5%) with third class New York Heart Association Functional Classification (53.0%) (NYHA) and an average diagnosis age of 61.92 (SD = 13.44). Most of the subjects, after being admitted to the clinic, underwent surgery (69.5%) for aortic valve replacement surgery, heart failure and myocardial revascularization. According to European guidelines, the first visit was made after 14 days. The main diagnoses were heart failure (35.9%) and coronary heart diseases (25%) followed by cardiomyopathies (17.2%) and mitral regurgitation (9.4%); some patients were hospitalized for post-surgical sequelae (12.5%). A large majority of patients were non-smokers or former smokers (87%) or suffered from hypertension (82%). There was a smaller percentage of patients suffering from other comorbidities such as diabetes (29.7%) and renal impairment (24.2%). Table 1 offers a detailed overview of patients’ clinical and demographic characteristics.

Our sample showed an average total score of 7.38 (SD = 3.09) out of 18 on the SAS subscale. Higher scores indicated a more intense experienced disorder. Self-reported ratings of QoL (SF-36) and psychosocial outcomes in terms of symptoms of anxiety and depression (HADS) are depicted in Table 1. The results indicated an average score of 32.0 (SD = 7.9) and 39.1 (SD = 10.2), respectively, for the physical health (PCS) and mental health (MCS) index of SF-36 (Short Form Health Survey 36). These scores showed a low perceived quality of life and deviated by nearly 2 SDs from the normative sample of the Italian population [26]. SF36 scores for gender are shown in Table 1. The comparison according to gender, median age and comorbidities examined showed no significant differences to the SAS scale. Patients with high scores on the HADS-A (*p* < 0.03) and HADS-D (*p* < 0.01) scales experienced most problems on the SAS scale (Table 1). Figure 1 showed experienced sexual problems based on gender, thus 21.3% of male patients and 12.8% of females indicated at least low or moderate intimacy reduction with their partners because of their condition, while the largest part of the sample reported no changes. About half of male participants reported from moderate to severe reduction of interest (49.5%), frequency (64%) and sexual activity limitations (41.3%). More than half the female participants reported a similar reduction from moderate to severe of interest (57.2%), frequency (69%) and sexual activity limitations (48.8%)

Correlations among the study variables were examined (Table 2). All the variables except SF36 mental health were significantly correlated with each other. SAS positively correlated with anxiety (r. 231, *p* = 0.009) and depression score (r. 0.215, *p* < 0.01) and negatively with SF36 physical health (r. −0.247, *p* = 0.005). SF36 physical health correlated negatively with anxiety (r. −0.261, *p* = 0.003) and depression score (r. −0.300, *p* < 0.001). Contrary to expectation, SF36 mental health was not correlated with SAS (r. −0.090, *p* = 0.314) while it was negatively correlated with anxiety (r. −0.397, *p* < 0.001) and depression (r. −0.579, *p* < 0.001).

### Mediation Analysis

In accordance with our hypotheses, we examined the relationship of the study variables by evaluating the mediating role of anxiety and depression. We applied a mediation model to analyze the total, direct and indirect effects of SF36 physical and mental health components on SAS through anxiety. The results indicated that only SF36 physical health is indirectly related to SAS through its relationship with anxiety. As shown in Figure 2, low scores on physical health are related to more anxiety (a = −133, *p* = 0.002), and higher anxiety was successively related to less sexual satisfaction (b = 0.143, *p* = 0.04). The 95% bias-corrected confidence interval based on 5000 bootstrap samples for the indirect effect (ab = −0.0191) did not include zero (−0.04352 to −0.00003), indicating an indirect effect [30]. A significant direct effect was detected. Specifically, the pathway where a lower physical health predicted sexual satisfaction was partially mediated by symptoms of anxiety.

To verify the path model with depression as a mediator we applied a mediation model to analyze the total, direct and indirect effects of SF36 physical health on SAS through depression. The results indicated that SF36 physical health is related to SAS but not through its relationship with depression. As shown in Figure 3, low scores on physical health are related to more depression (a = −173, *p* = 0.000). Unexpectedly, path b is not significant; indeed, depression was not related to SAS (b = 0.109, *p* = 0.08). Consequently, a 95% bias-corrected confidence interval based on 5,000 bootstrap samples for the indirect effect (ab = −0.04636) included zero (−11644 to 0.00513). Finally, a significant direct and indirect effect was detected.

## 4. Discussion

The main objective of this study was to investigate the mediating role of anxiety and depression in the relationship between quality of life and sexual satisfaction in CVD patients. The results from this study indicated that anxiety and not depression partially mediated the link among physical health and sexual satisfaction. These results could help to clarify the possible underlying relations between physical health QoL and sexual life in CVD patients. These patients experienced a significant level of physical, functional and emotional distress, showing a strong impact on health status. Our findings about the role of physical health on sexuality were consistent with those of a recent study highlighting that sexuality is closely linked to health in older people; moreover, physical health was associated with sexual problems more than age alone [31]. Our study confirmed the mediating role of anxiety, and depression was only associated with quality of life and not a mediator with sexual satisfaction. This result was in line with previous studies which found that both depression and anxiety played full roles in mediating between life satisfaction, gender differences, CVD, NYHA classification and quality of life [32,33,34,35]. Quality of life included an evaluation of physical health and mental health [25]. Patients with cardiovascular diseases are affected by lower physical and mental health, thus contributing to lower sexual satisfaction. This was contrary to a finding in a recent study [36] determining that mediation effects by depression status was not possible, probably due to the relatively small sample size and the fact that the cross-sectional data limited the ability to find causal relationships. This may also suggest a probable effect of one or more ignored mediators [37]. Conversely, anxiety but not depressive symptoms was linked to a stressful event such as a cardiovascular event [38]. In this study, low sexual satisfaction was shown to be associated with poor health outcomes (PCS and MCS). Indeed, sexual problems might have an impact on HRQoL; by contrast, a better HRQoL may also positively influence sexual relations [39]. There are several assumptions consistent with these results. The literature attributes great importance to the general health dimension, which includes both mental health and sexual health. These dimensions, in particular physical activity, are strongly related to sexual satisfaction in aging men and women [40].

According to our findings on the role of PCS, physical activity proved to improve cardiorespiratory functionality, strength and balance, cognitive functions, self-efficacy and depressive states [41]. Moreover, in the recent literature, good sexual activity as a form of physical activity is even more often related to mortality reduction, in both the general and the cardiac population, thus contributing to a better quality of life [42].

A chronic disease can affect sexuality in biological, psychological and social terms [43]. In reference to the theoretically based model, mediation analysis confirmed the interrelations between sexual satisfaction, mood status and quality of life in elderly patients with cardiovascular diseases. These conditions have a strong impact on different life aspects, including sexual functioning. Indeed, often sexual problems are common in both men and women cardiac patients and often are not adequately addressed in healthcare. However, changes in sex life are not always caused by the disease, but could be part of the natural aging process [44] and linked to normal hormonal changes, vascular tissue damage or reduction in muscular strength [45].

The results of our study also showed that specific dimensions of sexual relationships were affected by a cardiovascular disease in elderly patients. Specifically, about one male patient out of two reported a marked decrease in interest and 45.2% indicated a loss of sexual pleasure, while 57.2% of female patients reported a marked decrease in interest and 45.2% reported a loss of sexual pleasure. Most patients also reported a substantial reduction in frequency of sexual activity and felt hampered by the disease. In accordance with a study by Baert et al. [18], the results confirmed that if the sexual satisfaction dimension plays an important role, problems in sexual activity are very common in both genders and negatively affect patients’ quality of life and wellbeing [46].

However, if on one side healthier elderly people have reported better sexual satisfaction, the intimacy and interpersonal relationship dimensions proved to be less affected by cardiovascular disease. This result particularly reflects the complex relationship between mind and body and indicates that, in old age, when the sexual act is not desirable or possible anymore, the intimacy dimension becomes fundamental for emotional wellbeing. [47] Concerns about heart strain during sexual activity and physical difficulties could open up many other ways to express sexuality, such as close body contact, touching hands or kisses.

Our results showed that anxiety symptoms may affect sexual satisfaction for elderly CVD patients. Sexual satisfaction was significantly affected by the presence of anxiety. In previous research, depression and anxiety were related to decreased sexual arousal [48] and difficulties in communicating with one’s partner [49], which might lead to lower satisfaction within the sexual relationship.

In patients with CVD, anxiety may be negatively associated with sexual activity, thus worsening the couple’s sex life [50]. Specifically, performance anxiety is a factor influencing sexual health in patients with heart problems [51]. Moreover, anxiety and depression have been linked to lack of physical exertion [52] as patients who are anxious about their cardiovascular symptoms are more likely to avoid it. In addition, frequency of sexual activity is also a key mechanism through which sexual relationships may improve health outcomes [12] and may underline the influence of mood status on a decrease in sexual frequency. Indeed, as suggested by Mosack et al., decreased sexual activity after a cardiac event is related to anxiety or depression and a worse quality of life, adversely affecting psychological wellbeing [53].

Anxiety and depression may have a different influence on sexual satisfaction and our results underline which dimensions of SAS are most impacted by these states of mind. In patients with CVD, sexual satisfaction and frequency are impacted by anxiety and depression. These changes impair the patient’s quality of life and any intimate relationship, which can lead to anxiety or depression in a vicious circle [54]. Targeting mood symptoms may be an approach to breaking the vicious circle of sexual impairment and poor life quality in CVD patients, and help the health practitioner to pay more attention to subjective feelings of sexual wellbeing in connection with health [55].

Findings about gender differences in sexual satisfaction scores may also be ascribable to social factors and the living context. From a sociological point of view, it has been suggested that these gender differences may have been originated from the traditional gender roles and different social positions of genders [56]. There is no agreement in the literature about this. Indeed, some studies have reported that belonging to one gender or another may be a predictor of a worse quality of life, while other studies have not shown any gender association [57]. Nevertheless, the literature indicates that the most influential predictive factor of sexual wellbeing in aging is marital status, even if the most meaningful contribution to a decline of sexual satisfaction among women is the increase in the percentage of widows [5]. In this study, a worse quality of life (MCS and PCS scores) was related to anxiety and depression symptoms. Specifically, in our study, health deterioration makes participants aware of their age and this could lead to depression, anxiety states, Type D personality and sleeping disorders, causing psychosocial health deterioration [58]. All the factors we have underpinned might contribute to increasing a frail and vulnerable state in patients. For these reasons, it is important that healthcare professionals include a deep analysis of health and functional impairments in follow-up and routine treatments [59,60] as well as discussing patients’ sexual problems with them in a private setting within a wider context of psychological wellbeing. Despite great evidence of the role of anxiety and depression as major comorbidities in patients with CVD, they remain poorly recognized and inadequately treated. Increased attention to emotional and psychological wellbeing could improve outcomes over traditional effective therapies and increase the quality of life for these patients [61]. Psychologists and psychiatrists working in a cardiac rehabilitation context could be in the best position to assess and address these problems in the first rehabilitation phase, reassuring patients on the possibility of experiencing their sexuality in a different way, which may prevent the development of sexual problems and reduce sexual inactivity levels. Research on elderly sexual satisfaction and function will benefit from increased cultural sensitivity and interdisciplinary collaboration combined with more appropriate methodological approaches [5].

### Limitations

The results of the current study should be interpreted with caution, although they are supported by a recent literature analysis. The most important limit of the study is the relatively small sample size and the fact that it was recruited from only one clinical center. Our sample is made up of predominantly older men, and future research focused on the experience of younger people would be valuable. Another important aspect is that the answers to the questionnaires could be inaccurate. Self-reports of sexual behaviour and mood symptoms could be influenced by a number of factors including recall errors, social desirability and impression management [62].

Moreover, it was not possible to examine the impact of prescribed drugs because of both the small sample size and having to respect prescribed drug class uniformity in accordance with the European Cardiology Society guidelines. Furthermore, data on medical treatment were self-reported by the patients themselves, rather than obtained from medical records. Some of these clinical data appeared to be inaccurate, and other data were missing. Failure to adjust for these variables may have introduced a confounding bias. SF-36 is a generic quality of life instrument rather than a CVD-specific questionnaire. Finally, in this study a specific questionnaire was used for sexual satisfaction evaluation which showed high sensitivity to evaluating several aspects of sexuality.

Mediation models do not allow for causal inferences, suggesting a possible pathway of associations that should be investigated in future longitudinal or experimental studies.

However, other aspects such as the prevalence of a sexuality model based on reproduction, the impact of menopause and aging on women’s sexuality and the importance of social and cultural models that considered sexual activity as a privilege of young people, could affect the impact of aging in sexuality and for these reasons, they deserve more consideration in future studies.

## 5. Conclusions

This study investigated the underlying mechanisms of the association between sexual satisfaction and quality of life through a mediating model. Sexual satisfaction was shown to be negatively associated with physical health while depression and anxiety were negatively associated with physical health. Moreover, the findings indicated that anxiety but not depression played a mediating role in the relationship between sexual satisfaction and physical health. Experiencing sexual problems was meaningfully related to the physical health dimension. Overall, the findings suggest investigating perceived general health and the sexuality dimension as well as anxiety and depression when therapeutic decisions have to be made, given that they might be considered as clinical indicators to evaluate both risks for treatment and the best care for cardiovascular diseases.

## Figures and Tables

**Figure 1 healthcare-11-00290-f001:**
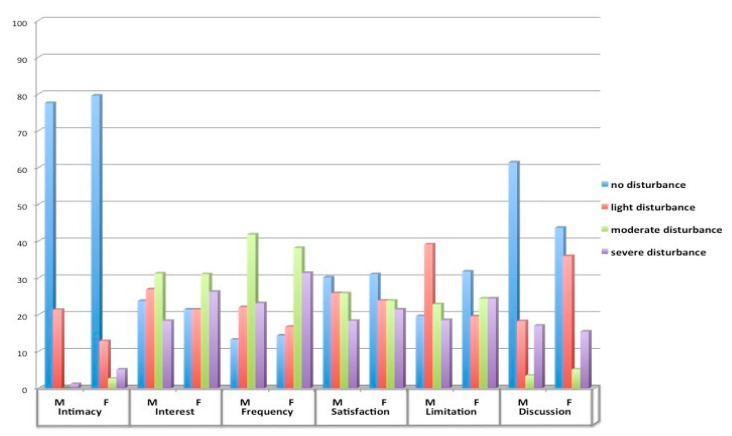
Sexual attitude scale (SAS) for gender.

**Figure 2 healthcare-11-00290-f002:**
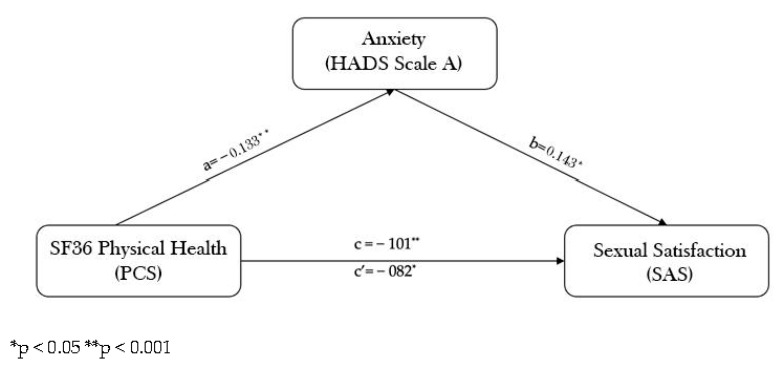
Mediating role of anxiety.

**Figure 3 healthcare-11-00290-f003:**
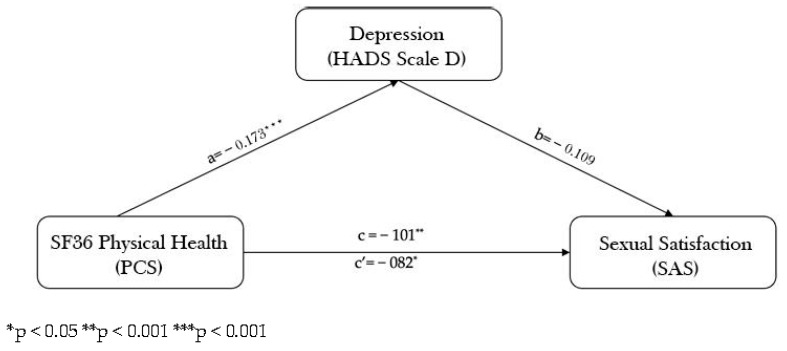
Mediating role of depression.

**Table 1 healthcare-11-00290-t001:** Characteristics of study population.

Characteristics (*N* = 128)		Sexual Adjustment Scale (SAS)		SF-36 (PCS)		SF-36 (MCS)	
	% (N) or Mean (SD)	Mean (SD)	*p*-Value	Mean (SD)	*p*-Value	Mean (SD)	*p*-Value
Gender							
Male Female	69.5% (89/128) 30.5% (39/128)	6.90 (3.37) 7.21 (3.62)	0.64	33.2 (7.82) 30.3 (7.77)	0.01	39.9 (10.7) 37.8 (9.23)	0.15
Age							
(Median)	<73 73–91	6.71 (3.52) 6.78 (3.74)	0.91	32.34 (8.03) 31.73 (7.80)	0.58	40.34 (10.39) 37.87 (9.81)	0.86
NYHA							
Class I Class II Class III Class IV	4% (5/128) 38% (49/128) 55% (70/128) 3% (4/128)	11 (0) 7.11 (3.22) 6.23 (3.51) 11 (0)	0.35	23.2 (3.5) 33.2 (8.54) 31.7 (7.23) 27.5 (3.54)	0.74	32.8 (5.06) 40.7 (10.1) 38.4 (9.39) 37 (7.07)	0.38
Diagnosis							
Cardiomyopathy Coronary heart disease Mitral regurgitation Heart failure Post-surgical sequelae	17.2% (22/128) 25% (32/128) 9.4% (12/128) 35.9% (46/128) 12.5% (16/128)	7.05 (2.89) 5.81 (3.68) 5.0 (3.63) 7.76 (3.26) 8.87 (2.50)	<0.01	30.2 (6.82) 33.4 (7.58) 34.6 (8.00) 31.5 (7.62) 30.1 (9.99	0.12	38.3 (12.2) 39.4 (10.3) 40.3 (8.82) 37.8 (9.43) 42.1 (10.8)	0.45
Intervention							
Yes No	69.5% (89/128) 30.5% (39/128)	6.93 (3.48) 7.13 (3.37)	0.76	32.2 (7.85) 31.7 (8.10)	0.68	39.3 (9.31) 38.6 (10.5)	0.64
Smoking History							
Yes No	13.3% (17/128)	6.2 (2.60) 7.10 (3.53)	0.34	31.9 (7.65) 32.0 (7.96)	0.93	37.1 (9.04) 39.4 (10.3)	0.29
Renal Failure							
Yes No	24.2% (31/128)	7.35 (2.75) 6.87 (3.64)	0.50	31.3 (7.54) 32.2 (8.02)	0.46	38.6 (8.04) 39.2 (10.7)	0.72
Diabetes							
Yes No	29.7% (38/128)	7.75 (2.76) 6.69 (3.64)	0.11	30.8 (8.34) 32.5 (7.7)	0.17	39.9 (8.81) 38.8 (10.6)	0.48
Hypertension							
	82% (105/128)	7.09 (3.19) 6.58 (4.38)	0.51	32.2 (7.99) 31.3 (7.54)	0.52	39.3 (9.61) 38 (12.4)	0.48
HADS-subscale A							
Normal Risk of distress Higher distress	6.09 (3.85)	6.45 (3.59) 7.24 (3.10) 8.78 (2.80)	0.03	33.5 (8.13) 31.3 (8.44) 28.2 (4.25)	<0.01	42.1 (9.87) 36.6 (9.26) 32.4 (8.18)	<0.001
HADS-subscale D							
Normal Risk of distress Higher distress	6.75 (4.35)	6.53 (3.63) 6.58 (3.03) 9.09 (2.83)	<0.01	34.7 (8.62) 31.3 (6.77) 27.4 (5.24)	<0.01	44.3 (9.52) 37.2 (8.17) 31.0 (7.66)	<0.001

The continuous variables are presented by mean (SD), while categorical variables are presented by valid hypertension (>140/90 mmHg), Sexual Adjustment Scale, Hospital Anxiety and Depression Scale, Quality of Life Evaluation Questionnaire (SF-36) %(N).

**Table 2 healthcare-11-00290-t002:** Pearson correlation between study variables.

Variables	1	2	3	4	5
1. SAS ^a^	1				
2. SF36 ^b^ (PCS)	−0.247 **	1			
3. SF36 ^b^ (MCS)	−0.090	0.140	1		
4. HADS ^c^ Scale A	0.231 **	−0.300 **	−0.397 **	1	
5. HADS ^c^ Scale D	0.215 *	−0.261 **	−0.579 **	0.541 **	1

* *p* < 0.05. ** *p* < 0.01. Sexual Adjustment Scaled ^a^, Quality of Life Evaluation Questionnaire (SF-36) ^b^, Hospital Anxiety and Depression Scale ^c^.

## Data Availability

Data are available upon request due to privacy restrictions.

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
