# Peer review of "Sexual Satisfaction and Quality of Life in Cardiovascular Patients: The Mediating Role of Anxiety"

_healthcare, 2023, doi:10.3390/healthcare11030290_

Round 1
Reviewer 1 Report
The authos has shown that sexual satisfaction (SAS) after CVD is negatively associated with SF-36 which is questionnares of HRQoL, conclude the anxiety is mediator relation between SAS and SF-36.
1. on the results, author shown SF-36 was not correlated with SAS (P=0.314)however, concluded that SF-36 was negatively associated with SAS in the conclusions. can you provide how to interpret this results?
2. Dose not matched significants between Table 2 and results. on results, SAS positively correlated with anxiety (r. .215, p = .01) and depression score (r. .215, 215 p <.01) but could not find a correlation on the Table 2. Authors demanding define of HADS-Scale A and B
3. Since authors has described in the limitation and results that differences by gender to evaluate sexual satisfaction, hereby, need to show differences/ gender in the table 1.
4. authors required intensively check spelling such as Sexual satisfaction evaluation (SAS) from introduction dose not matched Sexual adjustment scale (SAS) in the context, recommend to use same abbreviation.
5. there is no explanation to figure or graph between table1 and 2.
Author Response
We thank the Reviewer for her/his careful review and for her/his appreciation of our work.We fully agree with the Reviewer. In the new version of the manuscript we provided to:
- Correct and discuss the role of depression
- Correct table 2 and reported data in the text
- insert a new table to show a comparison according to gender, median age and comorbidities examined related to Sexual Satisfaction.
- Provide to a better define of SAS sub scale which examined the impact of the illness on patients’ sexual function and satisfaction.
- The legend was missing in Figure 1.
We have done our best to fully answer to her/his comments. We also provided a better discussion of limitation section. A native English teacher provided to the correction of the manuscript
Sincerly,
The Authors
Reviewer 2 Report
Dear Authors,
The topic is really interesting and indeed the sexual QoL of CVD patients is frequently overlooked in both studies and clinical practice. The description of the study you conducted is sufficiently detailed, my concerns being the following:
Line 133- English correction required - should be corrected the ,,they prostate problems''
Lines 161-162 - English rephrasing
Lines 337- English corrections required
You mentioned in your study that most of the patients underwent surgery after they were discharged from your rehabilitation facility. What type of surgeries did they have and could that influence their sexual QoL?
Do you performed an analysis of medication with the potential to affect sexual function? What part of the study group had such a chronic treatment? And what could have been the possible interference with your current results?
Kind regards.
Author Response
We thank the Reviewer for her/his careful review and for her/his appreciation of our work.We fully agree with the Reviewer. In the new version of the manuscript we provided to:
- A English rephrasing corrections required
- Most of the subjects, after being admitted to the clinic, underwent surgery (69.5%) for aortic valve replacement surgery, heart failure and myocardial revascularization. According to European guidelines, the first visit was made after 14 days. We ran a comparisons of study variable for surgery/not surgery to control influence on sexual satisfaction and quality of life. No significant differences were found. We attach here the table (not included in the manuscript)
|
Study Variables
|
Intervention |
No Intervention
|
p-Value |
|
Sas |
6.93 (3.48) |
7.13 (3.37) |
0.43 |
|
HADS-A |
6.20 (4.42) |
7.07 (7.06) |
0.18 |
|
HADS-D |
6.77 (4.41) |
8.05 (4.41) |
0.06 |
|
SF36 (PCS) |
32.2 (7.85) |
31.7 (8.10) |
0.68 |
|
SF36 (MCS) |
39.3 (10.5) |
38.6 (9.31) |
0.64 |
|
FE |
48.90 (14.46) |
43.58 (14.58) |
0.03 |
|
|
|
|
|
- We not performed an analysis of medication. Data on medical treatment were self-reported by the patients themselves, rather than taken from medical records. Some of these clinical data appeared to be inaccurate, others were missing. We discussed the possibility of a confounding bias in limitation section. However, in Table 1, we controlled Quality of life and Sexual satisfaction for comorbidities (related to the principal medications that could affect sexual life) to detect possible interference with our results. No differences were found.
We have done our best to fully answer to her/his comments. We also provided a better discussion of limitation section. A native English teacher provided to the correction of the manuscript
Sincerly,
The Authors
Reviewer 3 Report
Summary
Maria Stella Epifanio et al evaluated in a single centre study explored the underlying mechanisms of the association between sexual satisfaction and quality of life through a mediating model. I found this study interesting but some minor issues should be addressed.
Methods
· Statistics: In addition to the analyses performed, it would be interesting to perform a univariate and multivariate linear regression of the scores
Results
· Patients' medical therapy should be reported in the results and in Table 1
· It would be interesting to assess the medical therapy's influence on the scores.
Author Response
We thank the Reviewer for her/his careful review and for her/his appreciation of our work.We fully agree with the Reviewer. In the new version of the manuscript we provided to:
- A English rephrasing corrections required
- We not performed an analysis of medication. Data on medical treatment were self-reported by the patients themselves, rather than taken from medical records. Some of these clinical data appeared to be inaccurate, others were missing. We discussed the possibility of a confounding bias in limitation section.
- We proposed a new Table 1 where we controlled Quality of life and Sexual satisfaction for comorbidities (related to the principal medications that could affect sexual life) to detect possible interference with our results.
- Regressions model to detect significant path in Mediation model were showed in Figure 2-3.
We have done our best to fully answer to her/his comments
Sincerely,
The Authors
Round 2
Reviewer 2 Report
In my opinion is looking better and it should be published.
Author Response
Dear Reviewer,
We are grateful for your careful review and for the appreciation of our work.
Best regards
The authors